# Estimating Rotational Acceleration in Shoulder and Elbow Joints Using a Transformer Algorithm and a Fusion of Biosignals

**DOI:** 10.3390/s24061726

**Published:** 2024-03-07

**Authors:** Yu Bai, Xiaorong Guan, Long He, Zheng Wang, Zhong Li, Meng Zhu

**Affiliations:** 1School of Mechanical Engineering, Nanjing University of Science and Technology, Nanjing 210094, China; byu101010014@njust.edu.cn (Y.B.); wangzheng19@njust.edu.cn (Z.W.); zhong0814@njust.edu.cn (Z.L.); zhumeng0120@njust.edu.cn (M.Z.); 2Zhiyuan Research Institute, Hangzhou 310000, China

**Keywords:** mechanomyography, surface electromyography, transformer algorithm, estimation of human joint rotational acceleration

## Abstract

In the present study, we used a transformer model and a fusion of biosignals to estimate rotational acceleration in elbow and shoulder joints. To achieve our study objectives, we proposed a mechanomyography (MMG) signal isolation technique based on a variational mode decomposition (VMD) algorithm. Our results show that the VMD algorithm delivered excellent performance in MMG signal extraction compared to the commonly used technique of empirical mode decomposition (EMD). In addition, we found that transformer models delivered estimates of joint acceleration that were more precise than those produced by mainstream time series forecasting models. The average R^2^ values of transformer are 0.967, 0.968, and 0.935, respectively. Finally, we found that using a fusion of signals resulted in more precise estimation performance compared to using MMG signals alone. The differences between the average R^2^ values are 0.041, 0.053, and 0.043, respectively. Taken together, the VMD isolation method, the transformer algorithm and the signal fusion technique described in this paper can be seen as supplying a robust framework for estimating rotational acceleration in upper-limb joints. Further study is warranted to examine the effectiveness of this framework in other musculoskeletal contexts.

## 1. Introduction

In recent years, multimodal signal fusion methods and deep learning techniques for human activity recognition (HAR) have been extensively studied by researchers. Several such works may be considered for comparison with the study described in this paper. Dirgová Luptáková I et al. (2022) classified human activities using transformer models and achieved an accuracy rate of 99.2%. The device used for measurements in this research was a smartphone (accelerometer and gyroscope) placed in a bag on the waist of each person [1]. Wensel J et al. developed recurrent and vision transformer models to upgrade scalability and HAR speed; they obtained a level of accuracy that matched that of ResNet-LSTM but was achieved at twice the speed [2]. Shavit Y et al. (2021) evaluated a transformer model on diverse inertial datasets obtained over 27 h from 91 users, i.e., representing varying degrees of difficulty [3]. The scholarly literature also reveals the sheer number of biosignals that can be used for the recognition of human motion intention. Other recent studies have considered methods such as surface electromyography (sEMG), mechanomyography (MMG), and electroencephalography (EEG), as well as human–machine interactive forces [4,5].

A review of the literature reveals that one vital consideration when apply biosignals to estimate human motion is to ensure that wearing sensors will not affect human movement; this was previously emphasized by Zhang L et al. (2019) [6]. A specific trend in recent research has been to utilize MMG and sEMG for the estimation of joint acceleration, building on previous work on decoding movement via these signals. In comparison with more invasive or motion-constraining sensors, MMG and sEMG sensors are able to capture salient neuromuscular dynamics without imposing restrictions on users. Furthermore, the fusion of sEMG, which measures electrical activity driving muscle contraction, with MMG, which quantifies mechanical muscular vibrations, may result in the generation of complementary information. Further studies are warranted to determine optimal sensor arrangements and signal processing methodologies for the robust estimation of joint acceleration across diverse motion contexts.

A review of the literature also reveals the emerging use of transformer algorithms in the classification of human movement. Transformer models comprise a class of deep learning architectures that use attention mechanisms for processing sequential data, including pictures or natural language. Compared to recurrent neural networks (RNNs) or convolutional neural networks (CNNs), transformers offer several advantages when used for sequence modeling tasks. These include the following: (1) the ability to capture long-range dependencies between inputs, (2) efficient parallel calculation, and (3) state-of-the-art performance with respect to various natural language processing (NLP) benchmarks [7]. In this paper, we propose a new method for estimating rotational accelerations in elbow and shoulder joints using a fusion of biosignals and transformer algorithms. By using this method, we sought to harness the representation learning and sequence modeling capabilities of transformers to obtain the accurate decoding of limb dynamics from noninvasive neuromuscular signals. The usage of MMG and EMG enhanced the wearability of sensors without impeding movement. Using a simplified transformer model without a decoder part for motion estimation not only applied the attention mechanism of the transformer model but also simplified the structure of the motion estimation model and increased the operating efficiency and estimation accuracy.

In addition, we used the variational mode decomposition (VMD) algorithm for extracting MMG signals from raw acceleration measurements. A number of preparation techniques were then used to preprocess the acquired MMG and sEMG data. Ultimately, we established a transformer-based model for estimating joint acceleration using prepared MMG signals, sEMG signals, VMD-extracted pseudo-accelerations, and 3D camera-based movement tracker accelerations as model inputs.

## 2. Methods

This section begins with an overview of the experimental protocols and the sensor prototype used in the present study. The method proposed for MMG signal extraction is then described, along with the processing approaches applied to the acquired MMG and sEMG datasets. Finally, we describe the transformer model that was used for estimating joint acceleration in the present work. Transformers were originally developed for machine translation, but the more recent use of attention mechanisms by transformers has seen them emerge as an alternative to recurrent neural networks for sequence modeling, with the advantages of parallelization and faster training times.

### 2.1. Experimental Process and Sensor Prototype

The aim of the current experiment was to predict rotational accelerations in elbow and shoulder joints throughout three physical exercises: bicep curls, arm lateral raises, and arm frontal raises. Participants were instructed to execute movements at a frequency of their own choosing rather than at a prescribed rate. This was to promote naturalistic motion patterns that approximated daily activities. Informed consent was provided by 30 healthy adult participants (aged 24–27 years, including two females) after they were given the full details of the experimental protocols. The cohort size was adequate for a preliminary study; however, the number of participants should be increased in future studies so that statistical findings more closely and surely represent general populations.

The self-determined movements of participants enabled us to assess the ability of the models to generalize across movement speeds. To this end, for each physical exercise in the present study, 20 sets of 10 repetitions were completed by the participants, with 2–3 min of rest between sets to prevent fatigue. MMG and sEMG signals were acquired from the biceps brachii, anterior deltoid muscle, middle deltoid muscle, and posterior deltoid muscle using 4 special sensors. The arrangement of the sensors is shown in Figure 1a. Simultaneously, shoulder and elbow 3D rotational accelerations were recorded utilizing an optical gesture capture system. The prescribed movements were executed by participants upon instruction, and the data were collected. This level of physical exercise was sufficient for an initial validation. Greater robustness in the results might be obtained with more demanding exercise regimens; nevertheless, we believed that our preliminary research could offer indicative evidence of the proposed methodology’s utility and contribute to the search for a more rigorous experimental design.

The special sensor (Figure 1b) consisted of a 6-axis inertial measurement unit (MPU6050, TDK InvenSense, Sunnyvale, CA, USA), 2 surface electromyography electrodes with conductive gel, a STM32F103C8T6 microcontroller (ST, STMicroelectronics, Geneva, Switzerland), a nRF24L01 wireless module (NORDIC, Trondheim, Norway), and a lithium battery. The sensor was 5 cm long and 3 cm wide and weighed 98 g (including battery). It generated slight heat under long-term use but did not affect the progress of the experiment. Sensors were affixed over the target muscles using the conductive gel. The wireless design enabled untethered data streaming to a computer at 1000 Hz. Program for signal acquisition, processing, and motion estimation was implemented in Python on a PC (AMD Ryzen 5 5600X, 3.70 GHz). Sensor positioning and orientation were standardized to reduce placement-induced variability. Such a self-contained, wireless design offers a practical framework for unobtrusive monitoring outside laboratory settings. Further miniaturization and encapsulation could further enhance user comfort and robustness, enabling more prolonged periods of usage.

### 2.2. MMG Extraction and Biosignal Processing Method

#### 2.2.1. MMG Extraction Method

Mechanomyography (MMG) signals consist of low-frequency (10–50 Hz) muscular vibrations that are elicited throughout contractions. However, raw MMG recordings obtained using accelerometers contain high-frequency noise and motion artifacts. For the present study, then, signal decomposition was required to isolate the underlying MMG information. To this end, variational mode decomposition (VMD) was used. This involved an algorithm that separated a signal recursively into separate frequency–domain components by solving a constrained variational problem to minimize the mode bandwidths [8,9]. Compared to empirical mode decomposition (EMD), another method widely used for nonstationary signal analysis [10,11], VMD is characterized by greater sampling consistency and better noise robustness.

For the VMD-based MMG extraction method used in the present study, a fundamental preprocessing step was to isolate the 10–50 Hz mechanical vibrations from the raw sensor recordings. Subsequent analyses could then confirm the efficacy of VMD compared to alternatives such as EMD and also serve to characterize the effects of the tuning parameters on the preservation of balance and elimination of noise in relevant MMG signals.

The VMD algorithm had two fundamental parameters that required optimization: the penalty coefficient α and the mode number K. The number of intrinsic mode functions (IMFs) needed to decompose the signal into was determined by K. A too-low value of K raised the risk of insufficient separation and modal mixing; a too-high value of K raised the risk of identifying artificial components [12]. The bandwidths of the acquired IMFs were controlled by the penalty coefficient α, with lower values of α resulting in wider bandwidths. In previous studies, wider bandwidths have been associated with inter-mode leakage, while over-constrained bandwidths have been associated with loss of the informative signal content [13].

In light of the above, we applied differential evolution (DE), a population-based search algorithm, to adaptively tune the values of α and K. The effective use of DE begins with a population of candidate solutions. The solutions are subsequently improved using genetic operators like mutation, crossover, and selection [14]. By means of DE-based metaheuristic parameter optimization, we obtained a data-driven method to determine the penalty coefficient and VMD mode number. Use of this method removed any need for the exhaustive parameters required for tuning on a case-by-case basis.

Our application of differential evolution (DE) involved defining a fitness function to optimize the variational mode decomposition (VMD) parameters. A sparsity metric based on the uniformity of a signal’s probability distribution was provided by envelope entropy [15,16]. Inherent mode functions (IMFs) with higher levels of noise and fewer distinct components exhibited lower envelope entropy and greater sparsity. In contrast, IMFs with greater sparsity exhibited lower entropy. Therefore, envelope entropy was applied as the fitness criterion for VMD DE optimization, with the target of minimizing entropy so that maximally sparse signal decomposition was obtained. The envelope entropy E was defined as follows:(1)$$\begin{array}{c} E_{e}=-\sum_{j=1}^{N}\;e_{j}lge_{j}\\ e_{j}=a(j)/\sum_{j=1}^{N}\;a(j) \end{array}$$
where *E_e* is the envelope entropy, *e_j* is the normalized form of *a(j)*, *a(j)* is the envelope signal of the signal after Hilbert transformation, and N is the number of zero mean signals. The aim of this data-driven entropy minimization approach was to tune VMD to reject artifacts and isolate the MMG signal.

#### 2.2.2. MMG and sEMG Processing Method

MMG and sEMG signals are time series data; consequently, the preprocessing of both was required prior to the estimation of joint movement. The processing pipeline was as follows:(1)MMG extraction from raw accelerations using variational mode decomposition optimized by differential evolution (DE-VMD);(2)DC offset elimination for MMG and sEMG;(3)The 10–450 Hz bandpass filtering of both signals;(4)The full-wave rectification of both signals;(5)Linear envelope extraction for both signals;(6)The normalization of the processed signals.

The aim of this sequential pipeline was to isolate the characteristic MMG vibrations, remove noise and interference, smooth the data, and standardize the scales prior to input into the transformer architecture [17,18]. The entire preprocessing workflow is depicted in Figure 2.

Previous studies on wearable robotics have shown that system response delays in excess of 300 ms are perceivable and can impede normal human movement [19]. In the present study, we estimated offline movement; for any following real-time implementations, the preparation of continuous streaming data within this 300 ms was required. Specifically, it was necessary that signal acquisition, extraction, preprocessing, and model execution should all be completed prior to the next read cycle.

To this end, an input window duration of 1000 ms at increments of 200 ms was configured for the estimation model. This made sure that new estimates were available within the perceptual threshold; at the same time, continuity was ensured by the 800 ms overlap. Furthermore, the quantifications of data transmission and algorithmic contributions to the total delays enabled bottlenecks to be identified and addressed. The above methods were suitable for initial prototyping; however, the development of fully wearable hardware with optimizations for real-time usage is still required if fully practical human–robotic integration is to be achieved.

### 2.3. Transformer Model

The transformer architecture consisted of decoder and encoder components, as depicted in Figure 3. Both types of components used multi-head self-attention layers, pointwise feedforward layers, and fully connected layers. First, the input sequence was mapped by the encoder into a high-dimensional representation that encoded contextual interdependencies. The decoder then condensed this representation into the target output sequence, using masked self-attention to avoid information leakage. Modeling of long-range temporal relationships was enabled by attention in the data; nonlinear feature transformation was executed by the feedforward layers [20].

Encoder

As illustrated in Figure 3, the encoder consisted of stacked identical encoder layers. Each encoder layer was composed of two sublayers: a multi-head self-attention mechanism, followed by a position-wise feedforward neural network. The attention sublayer output was fed into the feedforward network, which used the same transformation at each sequence position. The two sublayers were connected with a residual connection and were then subjected to layer normalization.

2.Decoder

The decoder consisted of a stack of identical decoder layers, similar to the encoder just described. A self-attention mechanism followed by a feedforward network was also contained in each decoder layer. Additionally, encoder–decoder attention layers were inserted between pairs of sublayers.

3.Self-attention layer

a. Definition of attention

The definition of attention proposed by Google is as follows:(2)$$Attention(Q,K,V)=softmax\left(\frac{QK^{⊤}}{\sqrt{d_{k}}}\right)V$$
where *Q ∈ R^n×d^_k_*, *K ∈ R^m×d^_k_*, and *V ∈ R^m×d^_v_*. The attention layer encodes the sequence *Q* into a new sequence of *n × d_v_*.

b. Multi-head attention

An enhancement of the standard attention mechanism can be represented by multi-head attention in transformers. Formally, it projects the query (*Q*), key (*K*), and value (*V*) inputs through linear transformations to obtain h distinct representations of each.
(3)$${\mathrm{head}}_{i}=Attention\left(QW_{i}^{Q},KW_{i}^{K},VW_{i}^{V}\right)$$

Here, $W_{i}^{Q}\in R^{d_{k}\times {\tilde{d}}_{k}},W_{i}^{K}\in R^{d_{k}\times {\tilde{d}}_{k}},{and\;W}_{i}^{V}\in R^{d_{v}\times {\tilde{d}}_{v}}$, and then
(4)$$MultiHead(Q,K,V)=Concat({head}_{1},\ldots ,{head}_{h})$$

Finally, we obtain a sequence of $n\times \left(h{\tilde{d}}_{v}\right)$. The term ‘multi-head’ refers to doing the same thing repeatedly and then concatenating the results, so that
(5)Y=Multi–Head (X, X, X)

4.Position embedding

Position embeddings were applied to inject sequential-order information into the transformer model. All positions in the input and output sequences were assigned corresponding vectors via a lookup table. The formula for constructing position embedding may be expressed as follows:(6)$$\left\{\begin{array}{c} PE_{2i}(p)=sin\left(p/10000^{2i/d_{pos}}\right)\\ PE_{2i+1}(p)=cos\left(p/10000^{2i/d_{pos}}\right) \end{array}\right.$$

By such means, the position *p* is mapped to a *d_pos_* dimensional position vector, and the value of the *i*th element of this vector may be expressed as *PE_i_(p)*.

### 2.4. Application of a Transformer Model for the Estimation of Joint Acceleration

This model for estimating joint acceleration used the encoder part of the transformer for modeling only. A fully connected layer was used to project the tensor into the shape of [batch size, output length], and the output sequence was the estimating outcome of the input sequence [21]. The construction of the estimation model is illustrated in Figure 4. Application of the transformer-based model for the estimation of joint acceleration involved three key steps, as follows:Padding masks were utilized to pad all input sequences to a uniform length. Masking the added padding tokens to negative infinity enabled the softmax-normalized multi-head attention to ignore these positions effectively. This permitted variable-length inputs to be operated.Sinusoidal position encodings were injected into the input sequences. By assigning to each timestep a unique encoding based on sine/cosine functions at different frequencies, the model could generate sequential order information.The encoded input batch was passed through the encoder, consisting of a projection layer and encoder layers. Hierarchical features were extracted by the encoder layers via feedforward processing and self-attention. The last layer condensed the representation into the target output shape [batch size, output length]. The hyperparameters included batch size, attention heads, input sequence length, output length, number of encoder layers, and learning rate. All of these could be tuned by means of a grid search. After hyperparameter optimization, the best hyperparameter combination that was consistent with these experimental data was as follows: the batch size was 64, the number of input window/the number of output window was 20, the number of attention heads was 2, and the number of encoder layers was 1.

## 3. Results

In this section, we presented study results that demonstrated the efficacy of the variational mode decomposition (VMD) algorithm for MMG signal isolation and biosignal preprocessing pipelines. We then presented acceleration estimation results and compared these with results obtained using mainstream time series forecasting models. Finally, we reported the performance of the transformer model in different hyperparameters.

### 3.1. MMG Extraction and Biosignal Processing Results

DE-VMD was utilized to separate the pseudo-accelerations of joints and mechanomyography (MMG) signals from raw accelerometer data, as described above. Case decomposition results are shown in Figure 5 and Figure 6. The results for the original acceleration, pseudo-acceleration, and two IMFs, which are components of MMG, are depicted in Figure 5a. IMF1 was pseudo-acceleration, and IMF2 and IMF3 were components of MMG. The Marginal Hilbert Spectrum of corresponding components is shown in Figure 6a, which reflect the frequency–domain energy distribution. An examination of the figure revealed the effective isolation of the MMG signal (10–50 Hz) from higher-frequency noise and lower-frequency motion artifacts. The Marginal Hilbert Spectrum is a two-dimensional representation derived from the Hilbert spectrum; it shows the total energy contribution of each frequency value. The Marginal Hilbert Spectrum helped us understand the energy distribution across different frequencies in a signal, making it a valuable tool for various applications, including analyzing blood flow, climatic features, and water waves [22,23,24]. The efficacy of DE-VMD was compared to that of alternative methods such as empirical mode decomposition (EMD) with respect to information preservation and artifact rejection.

As previously mentioned, EMD is a method widely utilized for nonstationary signal decomposition. Unlike wavelet and Fourier transforms, which make use of predefined basis functions, EMD is data-driven and intrinsically adapts to signal characteristics. This delivers all the advantages associated with the nonlinear processing of nonstationary data with high noise levels. In the present study, for comparative purposes, EMD was applied to extract MMG signals, and the results are depicted in Figure 5 and Figure 6. The results for raw acceleration, extracted MMG, and low-frequency residue signal are displayed in Figure 5b. IMF1 and IMF2 were components of MMG, and IMF3 was a low-frequency residue signal. The Marginal Hilbert Spectrum of the corresponding components is displayed in Figure 6b. An examination of the figure showed that the EMD-derived MMG exhibits greater randomness and more dispersed spectral content in comparison with the DE-VMD extraction.

Figure 7 and Figure 8 illustrate the MMG and sEMG signals before and after the multistage preprocessing pipeline. It could be seen that raw nonstationary biosignals with high variability were transformed into smoother and more stationary signals after processing. This visual trend suggested the processing methods, including VMD extraction, filtering, rectification, and normalization, adequately prepared the sEMG data and MMG for application as inputs to the transformer-based model for estimating joint acceleration.

### 3.2. Estimation of Shoulder and Elbow Joint Acceleration Using a Fusion of Signals

To assess the efficacy of the proposed transformer-based method for the estimation of joint acceleration, comparative experiments were conducted using recurrent neural network (RNN) and long short-term memory (LSTM) architectures. LSTMs and RNNs might be seen as representative of mainstream sequence modeling techniques that are widely used today extensively for time series forecasting tasks.

Recurrent neural networks (RNNs) are a class of recursive architectures well-suited for sequential data, as previously stated. RNNs contain cyclical connections that enable the network state at each timestep to depend on prior context [25]. This provides a type of memory in contrast to feedforward networks, which lack an intrinsic temporal mechanism. Nevertheless, standard RNNs struggle to model longer-range dependencies because of vanishing/exploding gradients in the course of backpropagation over time. This limitation might be addressed using long short-term memory (LSTM) networks with gated-cell states, which allow for the persistence of information across extended sequences [25]. While RNNs are not able to link temporally distant context, longer-range patterns can be learnt by LSTMs. By comparing the proposed transformer model with LSTMs and RNNs, we sought to determine whether it would capture complex multimodal biosignal interdependencies more effectively and achieved more precise estimates of joint acceleration compared to the other two methods.

A comparison between the results for the recognition of joint acceleration obtained using the algorithm proposed here and those obtained using traditional algorithms is presented in Figure 9 and Table 1. In the figure, the blue curves are real joint acceleration signals of 3D movement experimentally captured, the yellow curves are estimation results of the transformer model, the gray curves are estimation results of the RNN model, and the brown curves are estimation results of the LSTM model. It can be seen that the transformer model produced the best estimates of joint acceleration, with the lowest rate of overall error among the algorithms, as shown by an estimation curve, which was smooth and without mutation.

### 3.3. Estimation of Shoulder and Elbow Joint Acceleration Using MMG Signals

A comparison between the results for joint acceleration recognition obtained using the algorithm newly proposed here and those obtained using traditional algorithms is presented in Figure 10 and Table 2. In the figure, the blue curves are real joint acceleration signals of 3D movement experimentally captured, the yellow curves are the estimation results of the transformer model, the gray curves are estimation results of the RNN model, and the brown curves are estimation results of the LSTM model. It can be seen that the transformer model produced the best estimates of joint acceleration, with the lowest rate of overall error among the algorithms.

### 3.4. Estimation of Shoulder and Elbow Joint Acceleration in Different Hyperparameters

Estimations in different hyperparameters are presented in Table 3. Because the number of features is eight, and the number of attention heads should be divisible by the number of features, so the number of attention heads are arranged as eight, four, two, and one, respectively. After finding the optimal number of attention heads, the optimal values of the other hyperparameters were also found through grid search. The estimation results in different numbers of attention heads are shown in Table 3(a). In addition, the estimation results in different numbers of encoder layers and different numbers of input windows/output windows are shown in Table 3(b,c). Among all kinds of hyperparameters, the number of attention heads, the number of encoder layers, and the number of input windows/output windows are most important, and the number of batch size does not have much impact on the estimation accuracy of the model.

## 4. Discussion

This study highlighted the following two advantages of the proposed methodology:(1)The use of transformer modeling capabilities.(2)The fusion of complementary multimodal biosignals.

Firstly, the results shown in Figure 9 and Figure 10 and Table 1 and Table 2 indicated that the precision of the proposed method was better than that of LSTM and RNN. The transformer’s capability to model complex spatiotemporal interdependencies within multimodal sequences was found to exceed the capabilities of LSTMs and RNNs, enabling a more precise inference of movement and further indicating that attention-based modeling captured the intricate relations between neural drives, muscle contractions, and limb accelerations more effectively than LSTM or RNN.

Secondly, the results shown in Table 4 indicated that precision was improved when joint movement was estimated using a fusion of EMG, MMG, and kinematic data compared to using MMG alone. This could be attributed to the heterogenous information conveyed by the different methods in the fusion. Because MMG measured mechanical vibrations and electrical potentials were detected by sEMG, the distinct insights provided by the two methods were combined into recruitment patterns and neuromuscular activation. Moreover, the signals were differentially influenced by factors such as fatigue and muscle length. Therefore, the aim of the fusion was to give a more complete representation of the motor unit behavior that drove movement. However, each constituent method had its own unique limitations, and these must be addressed as part of an effective fusion approach.

In addition, this study found that, due to the particularity of the experimental data, the number of enhanced layers and attention heads could not be too large; otherwise, it would lead to overfitting, which would lead to a decrease in accuracy. When the number of input windows/output windows increased, the accuracy of the model would increase. This was because, when the number of input windows/output windows increased, the model could use more historical data for predictions. However, the degree of increase in the accuracy would reduce as the number of input windows/output windows increased.

## 5. Conclusions

In this paper, we proposed a framework that combines differential evolution variational mode decomposition (DE-VMD) and transformer modeling to estimate acceleration in human shoulder and elbow joints. To be effective, any such estimation must address longstanding challenges posed by the inherent complexity of human movement, the ambiguity of sensing modalities, and noisy environments. The methodology proposed in the present study aimed to address factors related to both measurement complexities and body dynamics.

Comparative results indicated that the proposed method offered greater stability and superior precision in decoding upper-extremity movement than LSTM and RNN. The efficacy of the VMD method was validated by a systematic evaluation of MMG extraction. A fusion of MMG, sEMG, and kinematic data resulted in an enhanced context compared to a single signal. Ultimately, the capabilities of LSTM and RNN for representation learning were exceeded by the transformer when modeling the intricate spatiotemporal relationships that govern neuromuscular coordination.

The present study must be seen as preliminary in nature; nevertheless, the results reported here might serve to substantiate the utility of combining multimodal signal decomposition, fusion, and deep sequence modeling so that human movement intention might be inferred from wearable sensors. Further validation, involving more complex tasks and other limb joints, is warranted to determine the generalizability. However, the method described in this paper might be seen as a promising paradigm for further enhancing the robustness and precision that were essential for the recognition of human movement intention.

The present study might be seen as promising in its own right; however, several further avenues of research might also be motivated by the work reported in this paper. These include the following:

1. Enhancement of interpretability and model robustness. The current training data encompassed a limited, homogenous cohort performing simple tasks. Enlarging the diversity of subjects, biomechanics, and movements would enrich the dataset to improve generalizability.

2. Optimization of sensor configurations. The use of extra modalities (e.g., EEG), positions and sensor numbers, and on-body integration would help to determine an optimal balance of information gain and user load.

3. Investigation of complementary and alternative sequence models. A comparison of the transformer fusion method reported here with graphical methods and state-of-the-art convolutional techniques might reveal distinct advantages among different architectures.

4. Translation to online decoding on hardware. The introduction of self-contained and wearable platforms might be made possible by system optimization and miniaturization so that the recognition of movement intention can be achieved in real time.

In summary, the preliminary research described in this paper demonstrated the potential utility of a fusion sequence that modeled human movement intention. Further works involving different datasets, sensors, algorithms, and systems will be critical if these biosignals are to be used practically for the robust recognition of body movement.

## Figures and Tables

**Figure 1 sensors-24-01726-f001:**
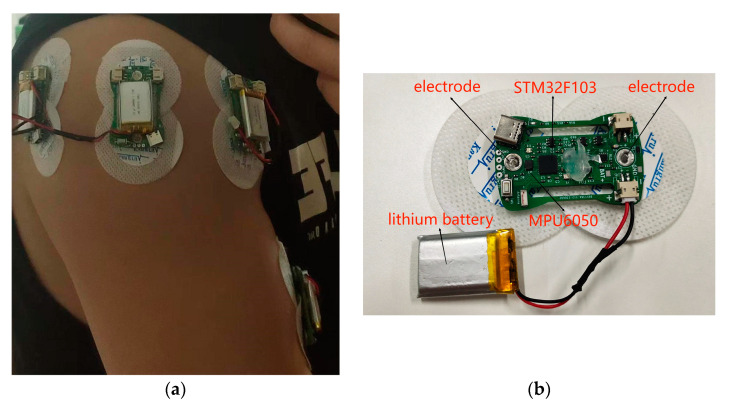
(**a**) The arrangement of sensors. (**b**) The composition of the special sensor.

**Figure 2 sensors-24-01726-f002:**
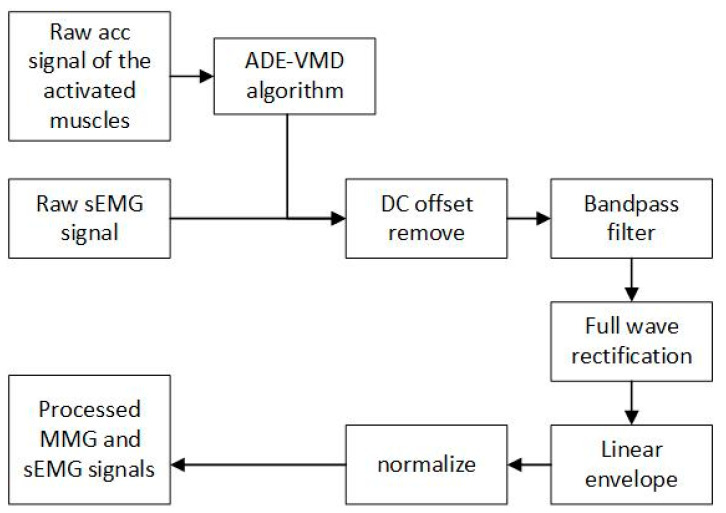
Multisource signal processing flowchart.

**Figure 3 sensors-24-01726-f003:**
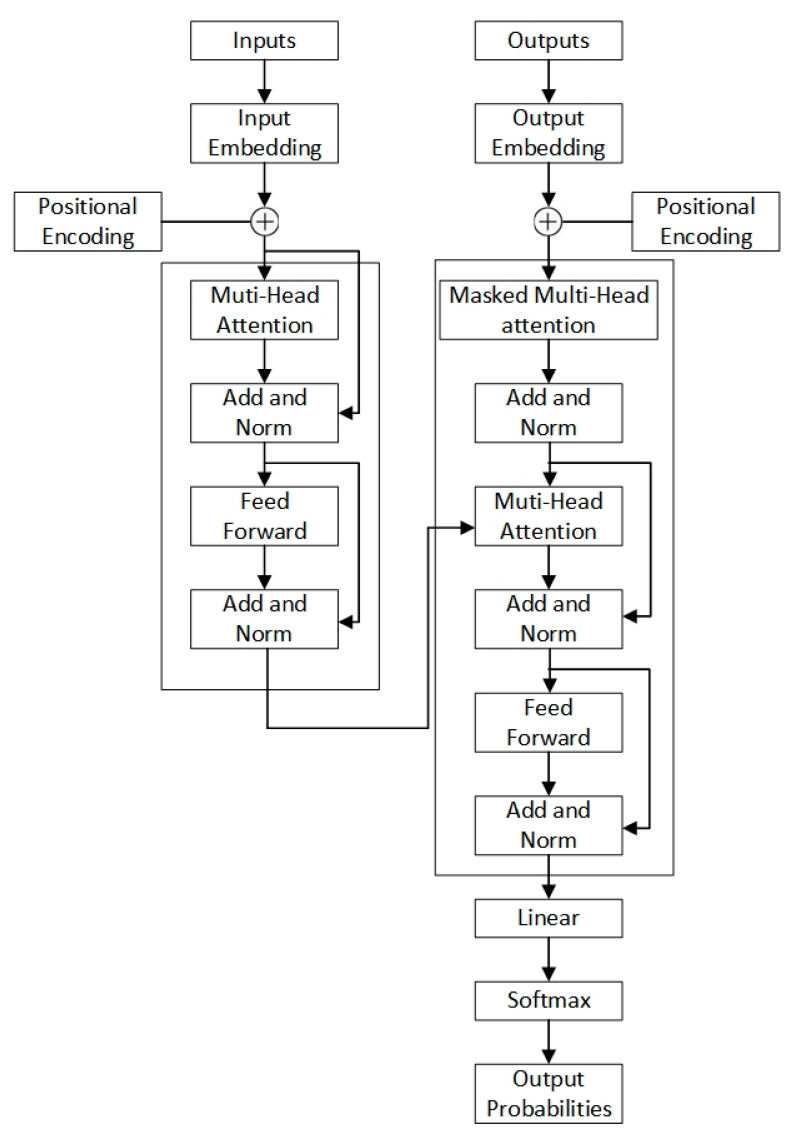
Transformer model.

**Figure 4 sensors-24-01726-f004:**
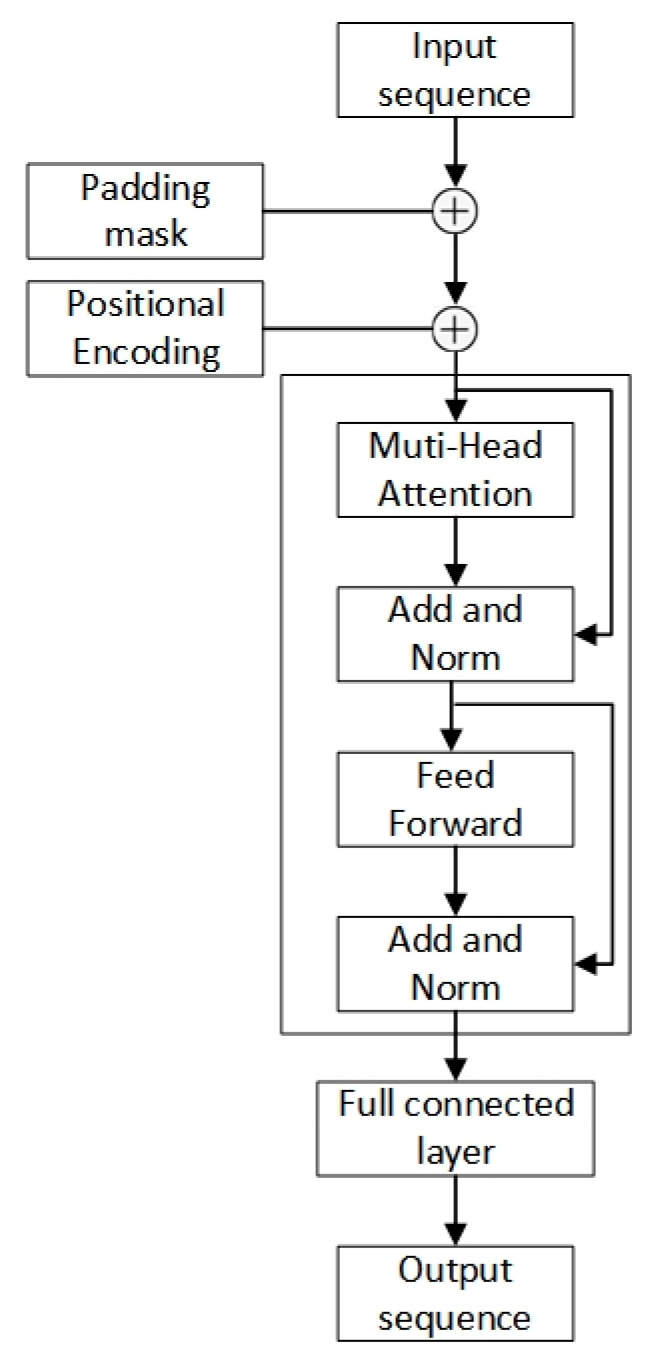
Transformer model for the estimation of joint acceleration.

**Figure 5 sensors-24-01726-f005:**
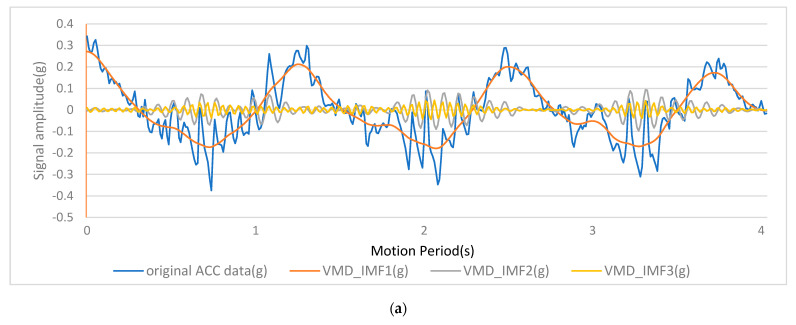
(**a**) Results of VMD in the time domain. (**b**) Results of EMD in the time domain.

**Figure 6 sensors-24-01726-f006:**
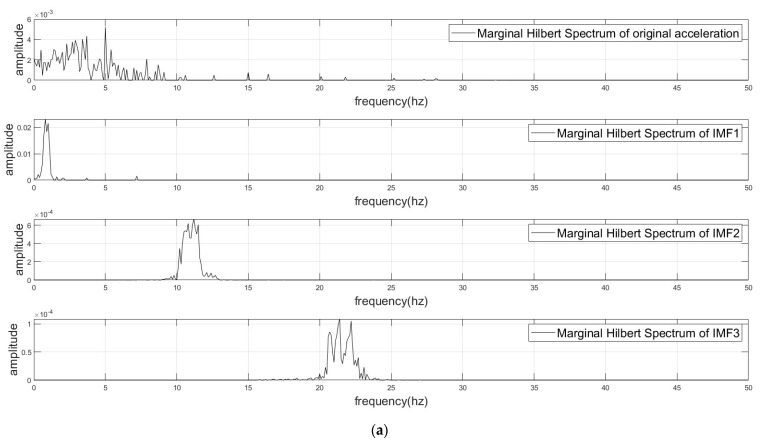
(**a**) Marginal Hilbert Spectrum of VMD. (**b**) Marginal Hilbert Spectrum of EMD.

**Figure 7 sensors-24-01726-f007:**
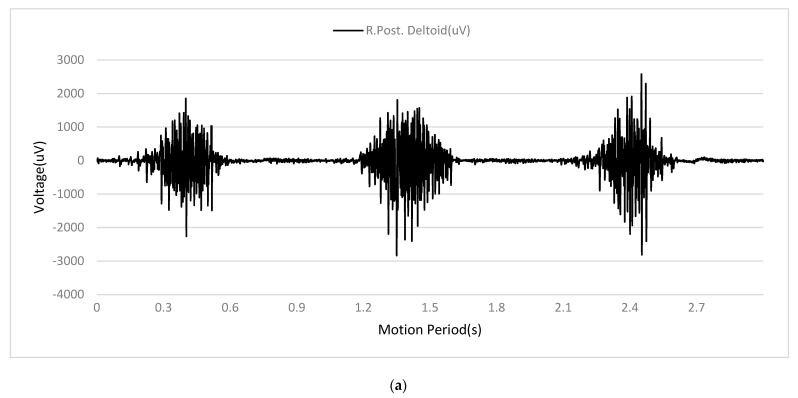
(**a**) Raw sEMG signal. (**b**) Raw MMG signal.

**Figure 8 sensors-24-01726-f008:**
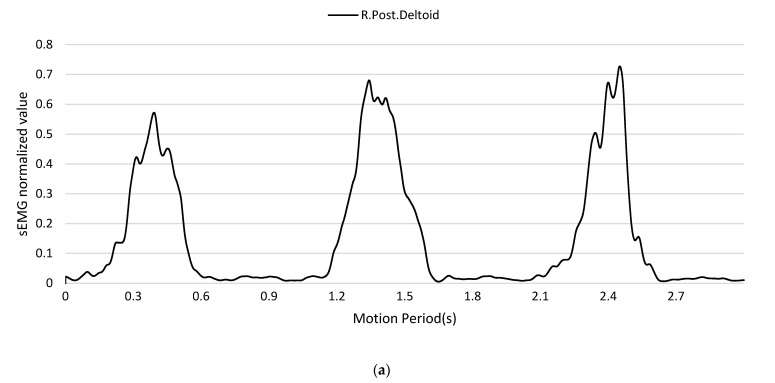
(**a**) Prepared sEMG signal. (**b**) Prepared MMG signal.

**Figure 9 sensors-24-01726-f009:**
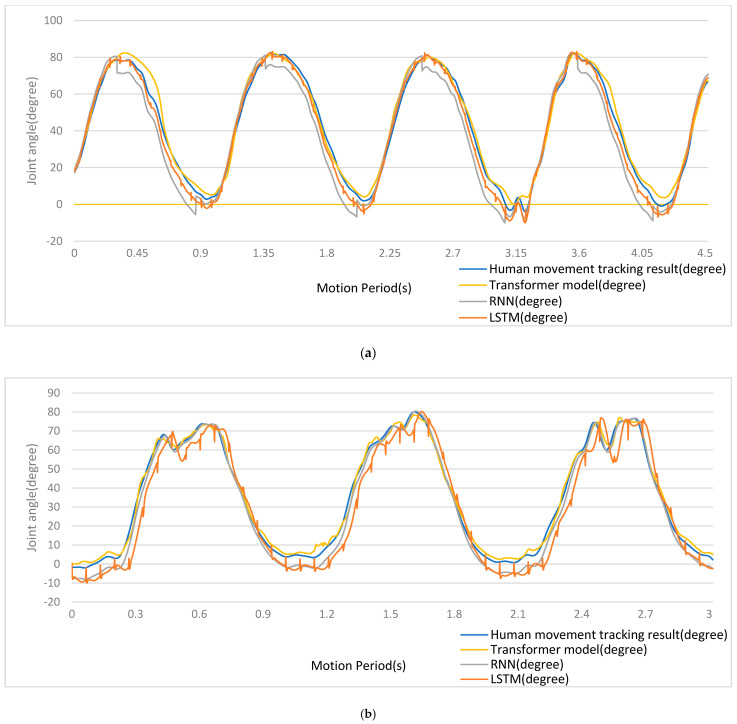
(**a**) Estimation results for arm side raises using a fusion of signals. (**b**) Estimation results for arm front raises using a fusion of signals. (**c**) Estimation results for elbow curls using a fusion of signals.

**Figure 10 sensors-24-01726-f010:**
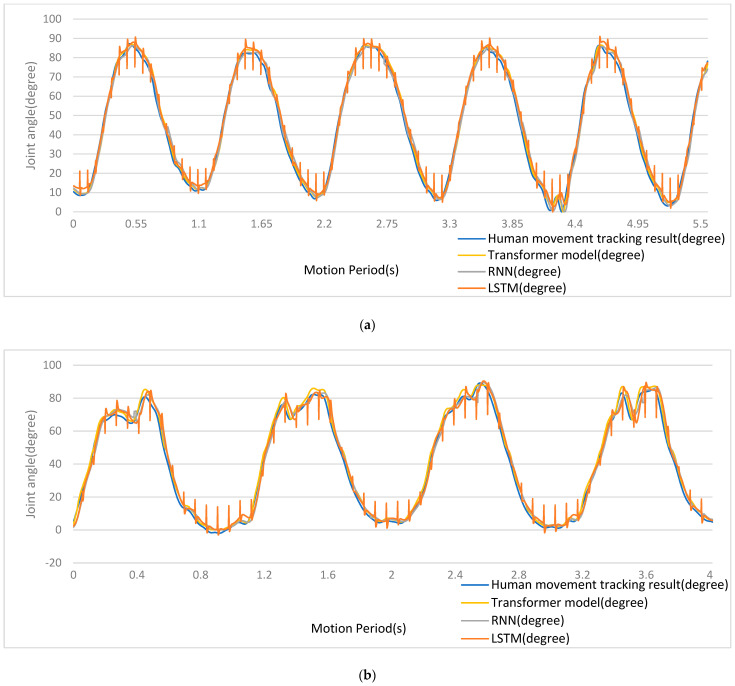
(**a**) Estimation results for arm side raises using MMG signals. (**b**) Estimation results for arm front raises using MMG signals. (**c**) Estimation results for elbow curls using MMG signals.

**Table 1 sensors-24-01726-t001:** (**a**) Estimation results for arm side raises using a fusion of signals. (**b**) Estimation results for arm front raises using a fusion of signals. (**c**) Estimation results for elbow curls using a fusion of signals.

**(a)**			
Algorithm	Highest Value of R^2^	Lowest Value of R^2^	Mean Value of R^2^
Transformer	0.976	0.942	0.967
RNN	0.956	0.926	0.941
LSTM	0.953	0.917	0.938
**(b)**			
Algorithm	Highest Value of R^2^	Lowest Value of R^2^	Mean Value of R^2^
Transformer	0.979	0.945	0.968
RNN	0.966	0.931	0.945
LSTM	0.961	0.922	0.939
**(c)**			
Algorithm	Highest Value of R^2^	Lowest Value of R^2^	Mean Value of R^2^
Transformer	0.957	0.901	0.935
RNN	0.915	0.852	0.889
LSTM	0.931	0.881	0.899

**Table 2 sensors-24-01726-t002:** (**a**) Estimation results for arm side raises using MMG signals. (**b**) Estimation results for arm front raises using MMG signals. (**c**) Estimation results for elbow curls using MMG signals.

**(a)**			
Algorithm	Highest Value of R^2^	Lowest Value of R^2^	Mean Value of R^2^
Transformer	0.952	0.902	0.926
RNN	0.923	0.861	0.897
LSTM	0.925	0.877	0.902
**(b)**			
Algorithm	Highest Value of R^2^	Lowest Value of R^2^	Mean Value of R^2^
Transformer	0.955	0.887	0.915
RNN	0.929	0.872	0.903
LSTM	0.926	0.874	0.888
**(c)**			
Algorithm	Highest Value of R^2^	Lowest Value of R^2^	Mean Value of R^2^
Transformer	0.931	0.852	0.892
RNN	0.867	0.826	0.841
LSTM	0.902	0.836	0.857

**Table 3 sensors-24-01726-t003:** (**a**) Estimation results for arm side raises in different numbers of attention heads. (**b**) Estimation results for arm side raises considering the number of encoder layers. (**c**) Estimation results for arm side raises in different numbers of input windows/output windows.

**(a)**			
the Number of Attention Heads	Highest Value of R^2^	Lowest Value of R^2^	Mean Value of R^2^
8	0.919	0.856	0.879
4	0.945	0.881	0.911
2	0.979	0.926	0.955
1	0.952	0.896	0.918
**(b)**			
the Number of Encoder Layers	Highest Value of R^2^	Lowest Value of R^2^	Mean Value of R^2^
20	0.611	0.552	0.585
15	0.761	0.726	0.745
10	0.875	0.851	0.863
5	0.941	0.876	0.915
2	0.97	0.931	0.953
1	0.979	0.926	0.955
**(c)**			
the Number of Input windows/Output Windows	Highest Value of R^2^	Lowest Value of R^2^	Mean Value of R^2^
5	0.741	0.618	0.667
10	0.802	0.704	0.752
15	0.895	0.829	0.854
20	0.979	0.926	0.955
40	0.978	0.927	0.955

**Table 4 sensors-24-01726-t004:** (**a**) Estimation results for arm side raises using the transformer model. (**b**) Estimation results for arm front raises using the transformer model. (**c**) Estimation results for elbow curls using the transformer model.

**(a)**			
Signal Type	Highest Value of R^2^	Lowest Value of R^2^	Mean Value of R^2^
Fusion signal	0.976	0.942	0.967
MMG signal	0.952	0.902	0.926
**(b)**			
Signal Type	Highest Value of R^2^	Lowest Value of R^2^	Mean Value of R^2^
Fusion signal	0.979	0.945	0.968
MMG signal	0.955	0.887	0.915
**(c)**			
Signal Type	Highest Value of R^2^	Lowest Value of R^2^	Mean Value of R^2^
Fusion signal	0.957	0.901	0.935
MMG signal	0.931	0.852	0.892

## Data Availability

Data are contained within the article.

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
