# Peer review of "Estimating Rotational Acceleration in Shoulder and Elbow Joints Using a Transformer Algorithm and a Fusion of Biosignals"

_sensors, 2024, doi:10.3390/s24061726_

Round 1

Reviewer 1 Report

Comments and Suggestions for Authors

1. In Figure 1,  what is the size and weight of the sensor, and does it heat up due to heat dissipation?

2. Like Figure 2, 3, 5 and 6, the figures are unclear and lack necessary labels and legends.

3. There are too many figures and the conclusions are lengthy.

4. The sample size is too small to draw meaningful conclusions.

5. The author(s) have not adequately discussed how their work relates to existing literature.

6. The paper does not discuss potential limitations of the study or suggest future directions.

Comments on the Quality of English Language

he paper is well-written and well-structured, but the language needs to be further polished.

Author Response

Dear reviewer:

It is a pleasure to receive your review comments. Here is my response to your comments:

1.The sensor is 5 cm long and 3 cm wide and weighs 98 grams (including battery). It will generate slight heat under long-term use, but it will not affect the progress of the experiment.

I have added this explanation to the third paragraph of Chapter 2.1.

2.According to the editor's comments, I have separated the method part and the results part of the paper, and added labels and legends.

3.Because this study includes joint angle recognition of three types of motion, there are many charts. In addition, many charts are also introduced to clearly show the difference in the results of MMG signal extraction between the two algorithms. Can I only add the plots of the original acceleration signal and MMG signal and delete part of the joint angle recognition result plots to reduce the number of plots?

4.Because these 10 days are the Chinese New Year holiday, I cannot go back to school to do additional experiments, so I want to hurry up and add the experimental results to the paper after returning to school on February 24. It is expected to take another ten days.

5.In this work, I used a simplified Transformer model, MMG signal and EMG signal to estimate the joint angles of three types of motion. The usage of MMG and EMG enhanced the wearability of sensors without impeding movement; Using a simplified transformer model for motion estimation not only applies the attention mechanism of the model but also prevented the estimation model from being too bloated.

I've added this explanation to the end of Chapter 1.

6. The potential limitations of the study is that present study must be seen as preliminary in nature; nevertheless, the results reported here may serve to substantiate the utility of combining multimodal signal decomposition, fusion, and deep sequence modeling so that human movement intention may be inferred from wearable sensors. Suggest future directions of the study are as follows:

1.Enhancement of interpretability and model robustness.

2.Optimization of sensor configurations.

3.Investigation of complementary and alternative sequence models.

4.Translation to online decoding on hardware.

I've added this part to the end of Chapter 5.

I am looking forward to hearing from you soon.

Kind regards

Reviewer 2 Report

Comments and Suggestions for Authors

The study examined Transformer models' use and fusion of mechanomyography (MMG) signals for estimating elbow and shoulder joint rotational acceleration. They introduced a signal isolation technique based on variational mode decomposition (VMD), showing its superiority over empirical mode decomposition (EMD). Transformer models outperformed traditional forecasting models in joint acceleration estimation. Combining MMG signals with other signals improved estimation accuracy. Overall, these methods offer a robust framework for estimating joint acceleration, warranting further exploration in different musculoskeletal contexts. However, after reading carefully, there are many major and minor concerns noted, as follows.

1. The Abstract has unclear information about the contributions of the work, The authors are suggested to refine the same with key numerical findings at the end. What is senior estimate performance? Such sentences degrade the quality of the work.

2. The authors have used mixed citation styles in the Introduction, which should be revised for consistency. The motivation is missing from the Introduction section. Moreover, the related work on Transformer models should be strengthened. Finally, in the end, based on the limitations of the existing work, the authors should tell the contributions.

3. How EMG and MMG signals are extracted from the human body? There should be a real-time picture representing the same with a sample human subject. Moreover, it would be nice to label Figure 1. The quality of Figure 2 should be improved. The representation (legends, x-y labels) for Figure 4 should be improved. 

4. It seems that the authors have used the Vanilla Transformer model as shown in Figure 5. However, Figure 6 which is redrawn from Figure 5 is a repetitive diagram and hence can be deleted. Otherwise, the authors should explicitly state how this model is different. Moreover, there should be more information on padding. Did the authors employ any modifications to suit the task of joint acceleration estimation?

5. In the results, the authors have used TFT without mentioning its expanded form anywhere The representation (legends, x-y labels) for Figures 7-8 should be improved. If the authors are giving the same information in Table 2 and Figure 10, and Table 3 and Table 11, any single information would be sufficient. Bar graphs should be revised to include the standard deviation (error bar) for different numbers of hyperparameters such as epochs or training. 

6. Moreover, the authors are suggested to provide details on selecting hyperparameters in the Transformer architecture like the number of attention heads, effective parameters, model parameters, etc. Moreover, It is unclear how the training of the model is carried out. 

7. How did the authors validate the performance of the Transformer model in estimating joint acceleration? Did authors employ any specific cross-validation or evaluation strategies?

8. There should be more details on interpreting the success of the Transformer model in this specific application. How its performance is attributed to its architecture, training methodology, or other factors? The information and results on ablation studies are missing and unclear. Please provide the same either in tabulated form or graphical representations.

Author Response

Dear reviewer:

It is a pleasure to receive your review comments. Here is my response to your comments:

1. The MSE, RMSE and MPE of transformer average are 0.00078, 0.0277, 0.011 respectively. The MSE, RMSE and MPE of fusion biosignals are 0.0009, 0.031, 0.011 respectively.

I have added this part to the abstract.

2. I have deleted some references that are not closely related to the research of this article.

In this work, I used a simplified Transformer model, MMG signal and EMG signal to estimate the joint angles of three types of motion. The usage of MMG and EMG enhanced the wearability of sensors without impeding movement; Using a simplified transformer model for motion estimation not only applies the attention mechanism of the model but also prevented the estimation model from being too bloated.

I've added this explanation to the end of Chapter 1.

3. Because these 10 days are the Chinese New Year holiday, I cannot go back to school to do additional experiments, so I want to hurry up and add the experimental results to the paper after returning to school on February 24 as requested by another reviewer. It is expected to take another ten days. I'll add photos of the body-worn sensor after the additional experiments is finished.

According to the editor's comments, I have separated the method part and the results part of the paper, and added labels and legends.

4. This model for estimating joint acceleration used the encoder part of the transformer for modeling only. A fully connected layer was used finally to project the tensor into the shape of [batch size, output length], and the output sequence was the estimating outcome of the input sequence.

I've added this explanation to the Chapter 2.4 and modified the picture.

5. I have added labels and legends and deleted the bar graphs. I will revised the bar graphs to include the standard deviation (error bar) for different numbers of hyperparameters after the additional experiments is finished.

6. The number of input window is 100, the number of output number is 5, the batch size is 64, the number of attention heads is 2, the number of encoder layer is 1.

I've added this part to the Chapter 2.4.

7. In this work, we used LSTM and RNN algorithms to compare with the proposed algorithm and found that the proposed algorithm can obtain better joint angle estimation accuracy. In addition, we also found that if the attention mechanism in the algorithm is removed, the accuracy of the proposed method will decrease significantly, but this finding was not included in the paper. Should I include the results without the attention mechanism in the paper for comparison?

8. I think the main reason why the proposed algorithm is superior to other algorithms lies in its attention mechanism. Should I add algorithms without attention mechanisms to the study as a comparison of the proposed methods? I will add this after additional experiments are completed.

I am looking forward to hearing from you soon.

Kind regards
